# Simulation and team training to improve preterm birth knowledge, evidence-based practices, and communication skills in midwives in Kenya and Uganda: Findings from a pre- and post-intervention analysis

**Lara Miller**[1]*, **Phillip Wanduru**[2], **Josline Wangia**[3], **Kimberly Calkins**[4], **Hilary Spindler**[1], **Elizabeth Butrick**[1], **Nicole Santos**[1], **Leah Kirumbi**[3], **Dilys Walker**[1]

**1** University of California, San Francisco Institute for Global Health Sciences, San Francisco, CA, United States of America, **2** Makerere University School of Public Health, Kampala, Uganda, **3** Kenya Medical Research Institute, Nairobi, Kenya, **4** PRONTO International, Seattle, WA, United States of America

* lara.miller@ucsf.edu, laralizmiller@gmail.com

**Data Availability Statement:** Data from this analysis available as a Dryad dataset: Miller, Lara (2023), PTBi EA simulation and team training –

## Abstract

Simulation training in basic and emergency obstetric and neonatal care has previously shown success in reducing maternal and neonatal mortality in low-resource settings. Though preterm birth is the leading cause of neonatal deaths, application of this training methodology geared specifically towards reducing preterm birth mortality and morbidity has not yet been implemented and evaluated. The East Africa Preterm Birth Initiative (PTBi-EA) was a multi-country cluster randomized controlled (CRCT) trial that successfully improved outcomes of preterm neonates in Migori County, Kenya and the Busoga region of Uganda through an intrapartum package of interventions. PRONTO simulation and team training (STT) was one component of this package and was introduced to maternity unit providers in 13 facilities. This analysis was nested within the larger CRCT and specifically looked at the impact of the STT portion of the intervention package. The PRONTO STT curriculum was modified to emphasize prematurity-related intrapartum and immediate postnatal care practices, such as assessment of gestational age, identification of preterm labour, and administration of antenatal corticosteroids. Knowledge and communication techniques were assessed at the beginning and end of the intervention through a multiple-choice knowledge test. Clinical skills and communication techniques used in context were assessed through the use of evidence-based practiced (EBPs) as documented in video-recorded simulations through StudioCode™ video analysis. Pre-and-post scores were compared in both categories using Chi-squared tests. Knowledge assessment scores improved from 51% to 73% with maternal-related questions improving from 61% to 74%, neonatal questions from 55% to 73%, and communication technique questions from 31% to 71%. The portion of indicated preterm birth EBPs performed in simulation increased from 55% to 80% with maternal-related EBPs improving from 48% to 73%, neonatal-related EBPs from 63% to 93%, and communication techniques from 52% to 69%. STT substantially increased preterm birth-specific knowledge and EBPs performed in simulation.

knowledge and skills assessment, Dryad, Dataset, https://doi.org/10.7272/Q61Z42PB.

**Funding:** The East Africa Preterm Birth Initiative was generously funded by the Bill and Melinda Gates Foundation (OPP1107312 to DW). The funders had no role in the study design, data collection and analysis, decision to publish, or preparation of the manuscript All authors received salary coverage through the grant to UCSF (LM, HS, EB, NS, DW) or sub-contracts with Makerere University School of Public Health (PW), the Kenya Medical Research Institute (JW and LK), or PRONTO Intentional (KC).

**Competing interests:** The authors have declared that no competing interests exist.

**Abbreviations:** ACS, antenatal corticosteroids; CRCT, cluster randomized controlled trial; EBPs, evidence-based practices; HBB, Helping Babies Breathe; IRB, institutional review board; LMICs, low- and middle-income countries; NG, nasogastric tube; NNR, neonatal resuscitation; PPV, positive pressure ventilation; PRONTO, PRONTO International; PTBi-EA, East Africa Preterm Birth Initiative; UCSF, University of California San Francisco; VA, video analyst.

# Background

In Kenya and Uganda, complications related to prematurity are the leading cause of neonatal deaths, therefore addressing the clinical care of mothers in preterm labour and babies born preterm is critical to reducing neonatal mortality rates [1–3]. Effective, low-cost, evidence-based practices (EBPs) have the potential to substantially reduce preterm-related mortality but their implementation requires not just facility-based delivery, but also skilled, knowledgeable providers. Modeling estimates suggest that strengthening the capacity of clinicians to deliver childbirth and post-childbirth services can avert a significant number of stillbirth and neonatal deaths, including provision of antenatal corticosteroids for preterm labour, management of preterm babies, and management of neonatal sepsis and pneumonia [4].

Effective provider training in intrapartum and neonatal skills, however, is notoriously challenging [5]. Neonatal resuscitation, for example, is critical to increase newborn survival, yet the skills are challenging to teach and retain, particularly in low-volume facilities with few opportunities for practice. The Helping Babies Breathe (HBB) curriculum developed by the American Pediatric Association in 2010 has been the most widely studied with a 2020 systematic review finding decreases in stillbirths and perinatal mortality across several studies, but a frequent decline in skills post-training without ongoing mentorship [6]. To achieve sustained improvement in newborn resuscitation, studies have shown that emphasis should be placed on mastery of skills, self-efficacy, and ongoing mentorship [7].

Simulation-based training, coupled with mentorship, has had success in both obstetric skill building and retention, as well as improved provider self-efficacy [8]. One such program, PRONTO International (PRONTO), designs curricula of low-cost, low-technology, simulation-based team training to improve basic and emergency obstetric and neonatal care specifically for use in limited-resource settings [9]. The PRONTO program builds upon the validated principles of simulation as a didactic tool by emphasizing EBPs, effective communication, teamwork, and respectful maternity care [10, 11]. Unlike the traditional Observed Structured Clinical Examination (OSCE) often used in Kenya and Uganda, in which examiners use a checklist to observe trainees performing EBPs, PRONTO uses video recording and allows examiners to later watch the videos and carefully identify any problems and provide feedback [12].

The East Africa Preterm Birth Initiative (PTBi-EA) implemented an intrapartum package of interventions aimed at improving knowledge and increasing the use of EBPs among facility-based maternity and neonatal care providers, with PRONTO simulation and team training (STT) as a key intervention component. The package included data strengthening, a modified version of the WHO Safe Childbirth Checklist, a quality improvement collaborative, and STT [13]. Through a cluster randomized controlled trial (CRCT) at 20 sites in Migori, Kenya and the Busoga region of Uganda, PTBi-EA demonstrated that the full intervention package resulted in a 34% reduced odds (odds ratio 0·66, 95% CI 0·54–0·81) of combined intrapartum and neonatal mortality among preterm infants compared to sites receiving a simplified package of data strengthening and the modified WHO Safe Childbirth Checklist alone [14]. To better understand how the STT trainings impacted preterm birth knowledge and skill of providers in the intervention facilities, we analysed knowledge test changes and EBPs as performed in simulation pre- and post-invention.

# Methods

## Study design

This study was a pre- and post-assessment of knowledge and EBPs performed in simulation embedded in the PTBi CRCT from 2016 to 2019. The protocol and results of the CRCT are published elsewhere [13, 14].

### Ethics approval and consent to participate

The IRB committees of the University of California, San Francisco (Study ID# 16–19162), the Kenya Medical Research Institute (Study ID# 0034/321), and Makerere University School of Public Health (Study ID# 189) reviewed and approved the PTBi-EA CRCT, including collection of process data related to each intervention component. We received approvals from the national and regional administrative leadership in each country to conduct this research and all participants in PRONTO training signed an informed consent form.

### Study population (sampling & inclusion/exclusion criteria)

Facility sites were chosen for the CRCT based on facilities that had 24-hour labor and delivery service, at least 200 births per year, and had a comparable facility to serve as a control. All thirteen of the study facilities participated in the STT and are included in this analysis. In Kenya, there was one regional referral hospital, five sub-county hospitals, two health centres, and one mission hospital. In Uganda, there was one regional referral hospital, two district hospitals, and one mission hospital.

Providers were selected based on their employment in the maternity or newborn wards of one of the intervention facilities, all clinicians (including medical officers, clinical officers, midwives, and nurses) in these two divisions were recruited into the study. An introductory session was held where all maternity and newborn ward clinicians were given an introduction to the intervention and asked to join the study. There were no rejections therefore all of the clinicians from these divisions received the intervention. All study participants signed informed consent documents.

### Description of the intervention–STT curriculum

PRONTO International worked in consultation with key stakeholders including representatives from Ministries of Health, regional clinical heads, and relevant professional organizations in each country and the PTBi-EA principal and co-investigators to adapt their curriculum to the PTBi-EA STT. The curriculum focused on identification, triage, and management of preterm labour, birth, and newborn care [15, 16]. The goal for consultations was to ensure that all clinical components reflected Ministry of Health guidelines from each country. The curriculum components included simulations, knowledge reviews and teamwork/communication activities. Further details of PRONTO's methodology and approach can be found elsewhere [9].

Several adaptations were made to the PRONTO program to emphasize preterm birth identification and management. First, the traditional PRONTO Pack supplies were expanded to include:

- A preterm-sized baby doll for realistic preterm labour and delivery

- A Nasco Life/form Micro-Preemie Simulator to practice more advanced preterm neonatal skills related to airways, chest tubes, gastrointestinal tracts, intravenous access, and nasogastric tube feeding [17]

- A Laerdal PreemieNatalie Preterm Simulator for preterm neonatal resuscitation [18]

- A Laerdal MamaBreast for practicing of breastfeeding and skin-to-skin [19]

- A neonatal weighing scale

- Simulated medications including vitamin K, tetracycline eye ointment, chlorhexidine, antenatal corticosteroids, magnesium sulfate (for neurodevelopment specifically)

Second, the PRONTO curriculum was expanded to include:

- Gestational age assessment knowledge reviews (calculation by menstrual history with standardized pregnancy wheels and fundal height measurements using tape measures)

- Modification of the simulation "pre-brief" to identify term versus preterm labour

- Integration of the PTBi-EA modified-WHO Safe Childbirth Checklist into simulation

- Greater emphasis on delayed cord clamping for prevention of neonatal anaemia for term and preterm babies

- Weight-based antibiotic dosing for septic babies

- Weight-based feeding protocols for preterm babies (including nasogastric tube feeding and intravenous glucose)

- Kangaroo care for low birthweight babies regardless of gestational age

- Administration protocols for simulated medications (see above)

- Inclusion of QI collaborative principles including, change ideas and plan, do, study, act cycles based on knowledge and/or skills gaps identified in simulation

## Training delivery and intervention dosage

In both countries, "PRONTO mentors" were locally recruited and trained in the PRONTO STT methodology and approach by PRONTO International. In Uganda, 10 mentors were recruited and trained including general physicians, specialist physicians, and nurse midwives. In Kenya, five mentors were recruited and trained all of whom were nurse midwives. All mentors in both countries received a 5-day simulation facilitator training in September 2016 led by PRONTO International covering curriculum, simulation facilitation, and debriefing skills and a 2-day refresher training in February 2017 (see Fig 1 for detailed timeline of activities).

The PRONTO mentors from each country implemented the STT intervention with the clinician-participants in their respective countries. Although the curriculum content was equivalent in both countries, Kenyan and Ugandan stakeholders chose two different models to implement the training [13]. In Kenya, two mentors visited each facility together for four consecutive days on 12 occasions during the three-year project (the "mentorship model"). During their visits, the mentors provided bedside mentorship, wherein they gave clinical guidance and feedback to healthcare workers in the antenatal, maternity, and neonatal wards. In between patients, they facilitated knowledge reviews, teamwork activities, and in-situ simulations and debriefs. In Uganda, the training was implemented in a modular fashion where visiting mentors trained healthcare workers at the facility over two consecutive days (the "modular model"). The training included facilitating knowledge reviews, skills stations, teamwork activities, and simulations in a classroom environment. Each facility in Uganda received 6 two-day trainings throughout the two years of the intervention, approximately one every three months. In addition to these visits, each facility in Uganda received five two-day bedside mentorship trainings in between curriculum training visits (Fig 1). Table 1 provides an overview of each country's intervention dosage.

## Data collection

**Knowledge assessments.** All participants were given a knowledge assessment test before the start and at the conclusion of the intervention. The two time points were: 1) October 2016

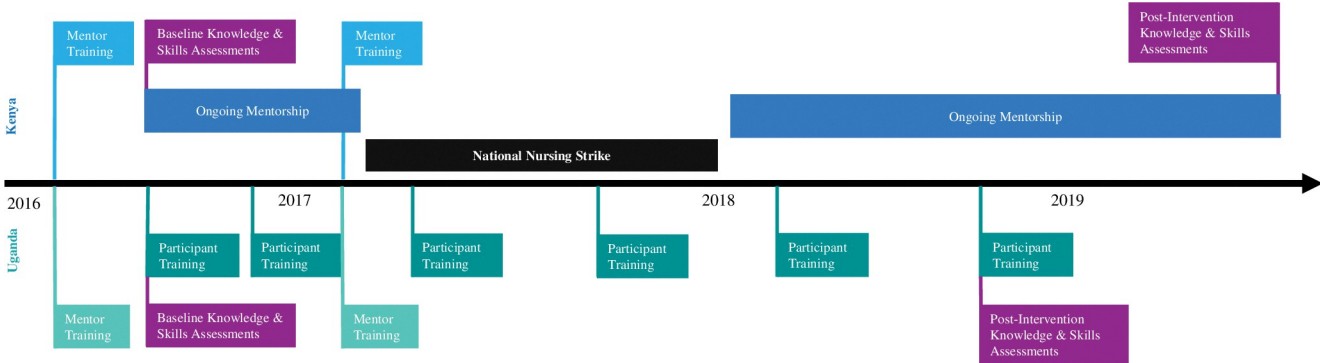

**Fig 1. Simulation & team training intervention timeline.**

(Kenya and Uganda), 2) August 2018 (Uganda) and April 2019 (Kenya) (Fig 1). Participants were given 45 minutes to complete the test at the beginning of a testing day. Only 22 of the 38 questions that related to preterm labour, preterm birth, care of the preterm newborn, and communication techniques (such as the two-challenge rule and thinking out loud) were used in this analysis. In both countries the pre- and post- intervention groups were not necessarily the same cohort of providers as some nurses may have transferred to other departments, some left or joined the facilities throughout the length of the intervention, or in Kenya the 7-month nursing strike resulted in significant staffing changes throughout their facilities.

**Skills assessments.** Participants' skills were assessed at the same baseline and conclusion timepoints described above. Only a subset of the participants were evaluated as only 2 participants acted as clinicians in each simulation. Three simulations dealt specifically with preterm birth and were included in this analysis. They included SimPack 4 (preterm premature rupture of membranes at 30 weeks with no delivery), SimPack 5 (preeclampsia at 36 weeks with delivery), and SimPack 6 (spontaneous vaginal delivery at 32 weeks with chorioamnionitis of a non-vigorous baby). Skills were assessed by the appropriate use of EBPs performed in a simulation (Table 2 for a decription of appropriate EBPs used in each in simulation).

During each training, the simulations were videorecorded and then shared with a trained video analyst (VA) in each country, a clinician who was not a PRONTO mentor. The VAs uploaded the videos into the software program StudioCode™ (version 5.8.4). StudioCode™ "windows" were developed in advance by clinical experts at UCSF, PRONTO International, KEMRI, and Makerere University to determine which EBPs should be performed in each specific simulation. The VAs watched the videos through the software and coded every time they saw an EBP performed. The data were then downloaded and transferred to the data analyst at UCSF. Three preterm birth-specific simulations were chosen to be evaluated at baseline and post-intervention (SimPacks 4–6) due to their emphasis on preterm labour, birth, and neonatal care and their repetition at multiple time points.

**Table 1. PRONTO curriculum components and average hours of exposure per facility.**

| | Number of curriculum components | Average hours of exposure per facility | |
| --- | --- | --- | --- |
| | | Kenya | Uganda |
| Knowledge Reviews & Skills Stations | 23 | 60 | 39 |
| Teamwork Activities | 11 | 14 | 21 |
| Simulations | 12 | 39 | 27 |
| Bedside Mentorship | - | 50 | 56 |

**Table 2. Evidence-based practices indicated in preterm simulations as part of the PRONTO simulation and team training.**

| Simulation description | SimPack4<br>Preterm premature rupture of membranes at 30 weeks (no delivery) | SimPack5<br>Preeclampsia at 36 weeks | SimPack6<br>Spontaneous preterm delivery at 32 weeks with chorioamnionitis of a floppy/non-vigorous baby |
|---|:---:|:---:|:---:|
| **Maternal clinical skills** | | | |
| Gestational age | ✓ | ✓ | ✓ |
| Fetal heartrate assessed | ✓ | ✓ | ✓ |
| Membranes evaluated | ✓ | ✓ | ✓ |
| Digital exam | | ✓ | ✓ |
| Correct diagnosis | ✓ | ✓ | ✓ |
| Antenatal corticosteroids | ✓ | | |
| Antibiotics | ✓ | | ✓ |
| **Neonatal clinical skills** | | | |
| Place on abdomen | | ✓ | ✓ |
| Stimulated and dried | | ✓ | ✓ |
| Suctioned | | ✓ | ✓ |
| Breathing check | | ✓ | ✓ |
| Head repositioned | | ✓ | ✓ |
| Positive pressure ventilation | | ✓ | ✓ |
| Chest rise | | ✓ | ✓ |
| Baby heartrate assessed | | ✓ | ✓ |
| Birthweight assessed | | ✓ | ✓ |
| Return skin to skin | | ✓ | ✓ |
| **Communication skills** | | | |
| Think out loud | ✓ | ✓ | ✓ |
| Check back | ✓ | ✓ | ✓ |
| Call for help | ✓ | ✓ | ✓ |
| SBAR (situation, background, assessment, and recommendation) | ✓ | ✓ | ✓ |
| Call out | ✓ | ✓ | ✓ |

**Data analysis.** Progress was evaluated not by individual performance changes, as individuals working in the facilities fluctuated, but by improvements at the intervention level for both changes in knowledge and EBPs performed in simulations.

Knowledge tests were entered into a Qualtrics form and coded for the correct answers. The data were analysed in RStudio version (1.0.13). Data were separated into 3 topics and comprised 7 maternal preterm labour and birth maternal questions, 10 preterm neonatal care questions, and 5 communication techniques questions. The proportion of correctly answered questions for baseline and intervention end were calculated for each question, and aggregated by topic (i.e., maternal, neonatal, and communication). Chi-squared tests compared the differences between aggregated data.

Data from StudioCode[TM] of the EBPs performed in simulation were also analysed in RStudio. EBPs were evaluated using a simple binary. The EBP was marked as "performed" if it was seen one or more times in the video of the simulation and "not performed" if it was not. Proportions of simulations where an indicated EBP was performed were calculated for baseline and intervention end. Maternal-related and neonatal-related EBPs were aggregated and compared using Chi-squared tests, in addition to overall proportions.

**Ethical approvals.** The IRB committees of the University of California, San Francisco (Study ID# 16–19162), the Kenya Medical Research Institute (Study ID# 0034/321), and

Makerere University School of Public Health (Study ID# 189) reviewed and approved the PTBi-EA CRCT, including collection of process data related to each intervention component. All participants in STT training signed an informed consent form. Participant information collected on the knowledge tests was limited to demographic data and is reported as country aggregates. In the simulation videos, the individual clinicians were identifiable to the VAs; however, the data shared with the UCSF data analyst were deidentified, and no identifying information was shared with any clinic or hospital administration.

**Inclusivity in global research.** Additional information regarding the ethical, cultural, and scientific considerations specific to inclusivity in global research is included in the Supporting Information as "Inclusivity in global research checklist."

## Results

There were 194 participants in Kenya and 124 in Uganda. Kenyan participants were predominantly female nurses, although there were significantly more male trainees (mostly clinical officers) than in Uganda where trainees were overwhelmingly female nurse midwives (Table 3).

### Changes in knowledge

Across all maternal, neonatal and communication questions, knowledge improved from 51.0% to 72.6% (p<0.01, Table 4). Four out of five questions related to maternal care improved with an overall increase of 13.1 percentage points, and 11 out of 12 neonatal care-related questions saw an improvement with an overall increase in 17.4 percentage points. The largest improvements in maternal questions were seen in questions related to preterm eclampsia management (+36.7 percentage points) and use of tocolytics (+23.0 percentage points). The largest improvements in neonatal-related questions were in neonatal resuscitation, specifically

**Table 3. Demographic data of simulation & team training participants.**

|  | Kenya | Uganda | Total |
|---|---|---|---|
| **Participants** |  |  |  |
| Total number | 194 | 124 | 318 |
| Number of facilities | 9 | 4 | 13 |
| Participants per facility (mean) | 22 | 31 | 53 |
| **Age** | **N = 185**[*] | **N = 115** | **300** |
| 18–29 | 36% | 30% | 33% |
| 30–39 | 37% | 32% | 35% |
| 40–49 | 19% | 27% | 23% |
| 50+ | 8% | 10% | 9% |
| **Profession** | **N = 194** | **N = 124** | **318** |
| Nurse midwife | 21% | 88% | 55% |
| Nurse | 45% | 2% | 24% |
| Clinical officer | 20% | 1% | 11% |
| Medical officer | 10% | 4% | 7% |
| Other | 4% | 5% | 5% |
| **Sex** | **N = 193** | **N = 119** | **312** |
| Female | 62% | 95% | 79% |
| Male | 38% | 1% | 20% |

*Differing Ns due to missing demographic data for certain participants.

**Table 4. Proportion of correct answers on a multiple-choice knowledge test administered at the beginning and end of the simulation & team training intervention.**

|  | Baseline (N = 318) | Post-Intervention (N = 233) | % Difference | p-value |
|---|---|---|---|---|
|  | % correct | % correct |  |  |
| **Preterm labour and birth maternal question topics** |  |  |  |  |
| Fundal height | 65.4 | 68.7 | 3.2 |  |
| Signs & symptoms of preterm labour | 26.2 | 18.5 | -7.8 |  |
| Use of antenatal corticosteroids | 87.0 | 97.4 | 10.4 |  |
| Use of tocolytics | 67.6 | 90.6 | 23.0 |  |
| Preterm eclampsia management | 57.7 | 94.4 | 36.7 |  |
| **All maternal questions** | **60.8** | **73.9** | **13.1** | **<0.01** |
| **Preterm & sick neonatal care question topics** |  |  |  |  |
| Preterm newborn care | 59.3 | 73.0 | 13.7 |  |
| Stable baby care (2.0kg) | 79.6 | 91.4 | 11.8 |  |
| Stable baby care (1.5kg) | 68.5 | 76.4 | 7.9 |  |
| CPAP indication | 19.8 | 13.3 | -6.4 |  |
| NNR: general | 71.3 | 90.6 | 19.3 |  |
| NNR: assessing pulse | 16.0 | 48.1 | 32.0 |  |
| NNR: assessing progress | 28.1 | 48.9 | 20.8 |  |
| NNR: assessing chest rise | 76.2 | 96.1 | 19.9 |  |
| NNR: with chest compressions | 44.4 | 72.5 | 28.1 |  |
| Sepsis symptoms | 71.9 | 94.4 | 22.5 |  |
| Sepsis management | 69.4 | 92.7 | 23.3 |  |
| Sepsis fluids | 57.7 | 73.4 | 15.7 |  |
| **All neonatal questions** | **55.2** | **72.6** | **17.4** | **<0.01** |
| **Communication questions** |  |  |  |  |
| "Check back" definition | 24.1 | 88.0 | 63.9 |  |
| "Shared mental model" definition | 25.9 | 42.1 | 16.1 |  |
| "Two-challenge rule" definition | 50.6 | 91.0 | 40.4 |  |
| SBAR example | 22.5 | 57.1 | 34.6 |  |
| General communication* | 33.3 | 77.7 | 44.3 |  |
| **All communication question** | **31.3** | **71.2** | **39.9** | **<0.01** |
| **TOTAL** | **51.0** | **72.6** | **21.5** | **<0.01** |

**Abbreviations**: CPAP–continuous positive air pressure, NNR–neonatal resuscitation, kg–kilograms, SBAR–situation, background, assessment, and recommendation.
*General communications questions included: thinking out loud, call outs, and other questions related to clear and concise clinical communication.

assessing pulse (+32.0 percentage points) and chest compressions (+28.1 percentage points). Signs and symptoms of preterm labour and CPAP indication were the two questions that saw decreases. Questions related to communication techniques improved by 39.9 percentage points with all questions showing improvement.

## EBPs performed during simulations

The proportion of preterm birth-related EBPs correctly performed in simulation saw an overall increase from 54.8% to 79.6% (p<0.01, Table 5). Maternal-related EBPs saw an improvement of 25.2 percentage points with the largest increases seen in the statement of correct diagnoses (+48.7 percentage points) and gestational age assessment (+28.6 percentage points). ACS, specifically, saw an increase of 19.4 percentage points. Neonatal-related EBPs saw an overall improvement from 63.0% to 93.1%, with the largest increases seen in neonatal assessments of heartrate and weight (+61.7 and +60.0 percentage points, respectively). The smallest

**Table 5. Evidence-based practices performed in simulation by a trainee at the beginning and end of the simulation & team training intervention.**

| | Baseline | | Post-Intervention | | Difference | |
|---|---|---|---|---|---|---|
| | No. of sims where EBP was indicated | % of Sims where EBP was performed | No. of sims where EBP was indicated | % of Sims where EBP was performed | % | p-value |
| **Maternal clinical skills** | | | | | | |
| Gestational age | 47* | 68.1 | 30* | 96.7 | +28.6 | |
| Fetal heartrate assessed | 47 | 61.7 | 30 | 83.3 | +21.6 | |
| Membranes evaluated | 47 | 23.4 | 30 | 40.0 | +16.6 | |
| Digital exam | 30 | 83.3 | 20 | 85.0 | +1.7 | |
| Correct diagnosis | 47 | 21.3 | 30 | 70.0 | +48.7 | |
| Antenatal corticosteroids | 17 | 70.6 | 10 | 90.0 | +19.4 | |
| Antibiotics | 39 | 33.3 | 19 | 57.9 | +24.6 | |
| **Sub-total** | **274** | **48.2** | **124** | **73.4** | **+25.2** | **<0.01** |
| **Neonatal clinical skills** | | | | | | |
| Place on abdomen | 30 | 100.0 | 20 | 100.0 | +0.0 | |
| Stimulated and dried | 30 | 83.3 | 20 | 100.0 | +16.7 | |
| Suctioned | 30 | 63.3 | 20 | 90.0 | +26.7 | |
| Breathing check | 30 | 16.7 | 20 | 65.0 | +48.3 | |
| Head repositioned | 30 | 90.0 | 20 | 95.0 | +5.0 | |
| PPV | 30 | 93.3 | 20 | 100.0 | +6.7 | |
| Chest rise | 30 | 76.7 | 20 | 100.0 | +23.3 | |
| Baby heartrate assessed | 30 | 23.3 | 20 | 85.0 | +61.7 | |
| Birthweight assessed | 30 | 40.0 | 20 | 100.0 | +60.0 | |
| Return skin to skin | 30 | 53.3 | 20 | 95.0 | +41.7 | |
| **Sub-total** | **300** | **63.0** | **200** | **93.0** | **+29.0** | **<0.01** |
| **Communication skills** | | | | | | |
| Think out loud | 47 | 70.2 | 30 | 93.3 | +23.1 | |
| Check back | 47 | 23.4 | 30 | 53.3 | +29.9 | |
| Call for help | 47 | 89.4 | 30 | 96.7 | +7.3 | |
| SBAR | 47 | 55.3 | 30 | 73.3 | +18.0 | |
| Call out | 47 | 21.3 | 30 | 26.7 | +5.4 | |
| **Sub-total** | **235** | **51.9** | **150** | **68.7** | **+16.8** | **<0.01** |
| **TOTAL** | **809** | **54.8** | **519** | **79.6** | **+24.4** | **<0.01** |

Abbreviations: EBP–evidence-based practice, PPV–positive pressure ventilation, SBAR–situation, background, assessment, and recommendation, Sim–Simulation

*Two clinicians participated in each simulation, for a total of 94 at baseline and 60 post-intervention.

improvements were seen in NNR head reposition management (+5.0 percentage points) and PPV (+6.7 percentage points) although both had high starting points of 90.0% and 93.3%, respectively. Communication techniques in simulation saw an increase of 16.8 percentage points with calling for help and thinking out loud both used in over 90% of simulations at 96.7% and 93.3%, respectively.

## Discussion

The PTBi-EA STT program of low-cost, low-technology simulation training, clinical mentorship, and emphasis on teamwork and communication showed an improvement in preterm labour and birth knowledge and EBPs performed in simulation over the course of the

intervention. Specifically, knowledge and practice of ACS, NNR, and sepsis management all improved, areas known to have a significant impact on improving preterm neonatal survival [4]. This finding lends support to the STT's contribution to the overall effect of the PTBi-EA intervention package, which saw a 34% decrease in combined fresh stillbirth and neonatal mortality in the intervention facilities compared to the control facilities [14]. As the PTBi-EA intervention package was designed to work synergistically, we cannot specifically separate the impact of the STT from the other components of the intervention package which also reinforced knowledge and EBPs. However, we believe that the combination of the four components resulted in an effect greater than the sum of its parts [20]. This finding is consistent with other combined simulation and ongoing mentorship programs which showed similar gains in knowledge and EBPs [21–26].

Recent publications from Kenya and Uganda highlight the ongoing need for the improved skills and knowledge of maternity and neonatal care providers [27, 28]. Ongoing HBB programs have shown success particularly when video debrief was added to the skills training curriculum [26]. Additionally, the creation of newborn care units in facilities have shown significant improvements in neonatal outcomes in a program in Eastern Uganda highlighting the impact of a low-resource, but targeted effort to improve conditions [29].

Many recent studies have shown the need for improvement in clinical maternity and newborn care in Kenya and Uganda, including a 2021 review of Ugandan midwifery training programs which concluded that they are too numerous and too disparate without clear pathways for advancements and did not meet the International Confederation of Midwives standard [27]. A recent modeling study by Nove et. al found that an increase in midwifery care and midwifery-delivered EBPs could avert 39% of neonatal deaths and 26% of stillbirths in low- and middle-income countries, but the study cautioned that midwifery care alone was insufficient and stressed the importance of sustained improvement in the knowledge and skills of midwives [28].

An evaluation of a high-tech simulation program in a hospital in Kampala, Uganda found substantial increase in knowledge after the training but fewer improvements in medical skills and no change in teamwork [30]. A 2019 evaluation of a clinical mentorship program in emergency obstetrics and newborn care (EMONC) in rural Kenya, saw a significant decrease in stillbirths and an increase in appropriate referrals for higher level care, concluding that targeted mentorship was a key component of sustained improvement in nurse/midwife skills [31]. A HBB evaluation across several countries including Kenya found that although both knowledge and skills improved directly after training, skills declined over time and required ongoing practices and mentorship was needed [32]. A separate evaluation of an HBB program in Uganda found that the addition of video debriefing improved retention and skills competency when evaluated 6-months post-training [26]. These recent studies highlight the need for teamwork and communication, sustained mentorship, and video debriefs, key components of the PTBi-EA STT program.

While PRONTO STT training is not the only approach to improve preterm knowledge and skills in low-resource healthcare facilities, it is a model that has proved effective in improving maternal newborn, and now preterm specific newborn outcomes [33, 34].

## Limitations

There are several limitations to this study. First, it is not known whether knowledge or skills performed in simulation translates into performance in live clinical scenarios. Capturing skills performed during actual birth is difficult and expensive, requiring in-person or video surveillance. However, based on the results of the PTBi-EA CRCT, it is likely that maternal and

neonatal skills did increase in actual clinical practice. Second, pre- and post- knowledge test scores have disparate sample sizes and are presented as aggregated data per timepoint, rather than as changes on an individual level. While this limits analysis (e.g., evaluating if different cadres of providers differentially benefited from the training), it emphasizes the PRONTO approach of focusing on the clinical team, rather than the individual provider. While this was the design of the evaluation program it did limit assessments of individual providers. Third, during the CRCT, Kenya had a national physician and nursing strikes intermittently between December 2016 and October 2017, with the nurses striking for six consecutive months between May and October 2017 [35]. During that time, intervention activities halted, including clinical mentorship. When activities began again in November 2017, many of the initial training cohort had left or were transferred to other facilities, so much of the PRONTO curriculum needed to be repeated for new trainees. Additionally, as mentioned above, the in-situ mentorship model meant there were fewer trainees present for the post-intervention knowledge test than the baseline test which resulted in disparate denominators. Despite this, the fact that significant improvements were still seen in Kenya over the 30-month intervention period speaks to the strength of the PTBi-EA intervention package as a whole and the reinforcement of best practices through quality improvement work and the WHO Safe Childbirth Checklist.

## Conclusion

This study shows that the PTBi-EA STT is efficacious in improving provider knowledge and skills related to preterm labour, birth, and neonatal care. Improving the care of mothers in preterm labour and babies born preterm is critical to reducing the rates of neonatal mortality in East Africa and elsewhere in the world. An intrapartum quality improvement package, with simulation and team training at its core, is an approach that other contexts may consider to improve survival of preterm babies.

## Acknowledgments

The authors gratefully acknowledge the providers who participated in the PRONTO trainings, PRONTO mentors and consultants. We thank hospital administration and local health authorities in Migori County, Kenya and Busoga region, Uganda for their partnership in this work.

## Author Contributions

**Conceptualization:** Lara Miller, Phillip Wanduru, Hilary Spindler, Elizabeth Butrick, Nicole Santos, Leah Kirumbi, Dilys Walker.

**Data curation:** Lara Miller, Phillip Wanduru, Josline Wangia, Kimberly Calkins, Hilary Spindler, Elizabeth Butrick, Nicole Santos.

**Formal analysis:** Lara Miller, Phillip Wanduru, Josline Wangia, Hilary Spindler, Dilys Walker.

**Funding acquisition:** Elizabeth Butrick, Nicole Santos, Dilys Walker.

**Investigation:** Lara Miller, Phillip Wanduru, Josline Wangia, Kimberly Calkins, Hilary Spindler, Elizabeth Butrick, Nicole Santos, Leah Kirumbi, Dilys Walker.

**Methodology:** Lara Miller, Kimberly Calkins, Elizabeth Butrick, Nicole Santos.

**Project administration:** Lara Miller, Phillip Wanduru, Josline Wangia, Kimberly Calkins, Elizabeth Butrick, Leah Kirumbi, Dilys Walker.

**Resources:** Elizabeth Butrick, Dilys Walker.

**Software:** Hilary Spindler.

**Supervision:** Elizabeth Butrick, Leah Kirumbi, Dilys Walker.

**Writing – original draft:** Lara Miller, Phillip Wanduru, Josline Wangia.

**Writing – review & editing:** Lara Miller, Kimberly Calkins, Hilary Spindler, Elizabeth Butrick, Nicole Santos, Leah Kirumbi, Dilys Walker.

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
