## [Decision Letter · Decision Letter 0]

4 Jan 2023

PGPH-D-22-01386

Simulation and team training to improve preterm birth knowledge, evidence-based practices, and communication skills in midwives in Kenya and Uganda

Dear Dr. Miller,

Thank you for submitting your manuscript to PLOS Global Public Health. After careful consideration, we feel that it has merit but does not fully meet PLOS Global Public Health’s publication criteria as it currently stands. Therefore, we invite you to submit a revised version of the manuscript that addresses the points raised during the review process (two reviewers have provided constructive feedback below).

We look forward to receiving your revised manuscript.

Kind regards,

Hannah Tappis, DrPH, MPH

Academic Editor

Journal Requirements:

2. Please send a completed 'Competing Interests' statement, including any COIs declared by your co-authors. If you have no competing interests to declare, please state "The authors have declared that no competing interests exist". Otherwise please declare all competing interests beginning with the statement "I have read the journal's policy and the authors of this manuscript have the following competing interests:"

3. Please amend your detailed Financial Disclosure statement. This is published with the article. It must therefore be completed in full sentences and contain the exact wording you wish to be published.

a. Please clarify all sources of funding (financial or material support) for your study. List the grants (with grant number) or organizations (with url) that supported your study, including funding received from your institution. 

b. State the initials, alongside each funding source, of each author to receive each grant.

c. State what role the funders took in the study. If the funders had no role in your study, please state: “The funders had no role in study design, data collection and analysis, decision to publish, or preparation of the manuscript.”

d. If any authors received a salary from any of your funders, please state which authors and which funders.

4. We do not publish any copyright or trademark symbols that usually accompany proprietary names, eg (R), (C), or TM  (e.g. next to drug or reagent names). Please remove all instances of trademark/copyright symbols throughout the text, including R on page 9 and TM on page 6, 9, 10 and 11.

5. In the online submission form, you indicated that your data will be submitted to the Dryad database upon acceptance. Should your submission be accepted, we will require the following information in your Data Availability Statement: 

a. The DOI provided by Dryad

b. The citation for your data package in the reference section of your manuscript

c. The citation for your data package in the methods section

If you are unable to adhere to our open data policy, please kindly revise your statement to explain your reasoning and we will seek the editor's input on an exemption. Please be assured that, once you have provided your new statement, the assessment of your exemption will not hold up the peer review process.

Additional Editor Comments (if provided):

Reviewers' comments:

Reviewer's Responses to Questions

**Comments to the Author**

1. Does this manuscript meet PLOS Global Public Health’s publication criteria? Is the manuscript technically sound, and do the data support the conclusions? The manuscript must describe methodologically and ethically rigorous research with conclusions that are appropriately drawn based on the data presented.

Reviewer #1: Yes

Reviewer #2: Partly

2. Has the statistical analysis been performed appropriately and rigorously?

Reviewer #1: Yes

Reviewer #2: I don't know

3. Have the authors made all data underlying the findings in their manuscript fully available (please refer to the Data Availability Statement at the start of the manuscript PDF file)?

Reviewer #1: Yes

Reviewer #2: Yes

4. Is the manuscript presented in an intelligible fashion and written in standard English?

Reviewer #1: Yes

Reviewer #2: Yes

5. Review Comments to the Author

Reviewer #1: Thanks for inviting me to review this paper which reports the results of a simulation and training package to improve preterm birth knowledge, evidence-based practices and communication skills. I thoroughly enjoyed reading the manuscript: it is very well written, and I have only very minor comments. Congratulations to the authors for conducting this study. The biggest limitation is clearly that it is not possible to extrapolate the impact that PRONTO specifically had, given that it was part of a wider programme, however, the authors recognise this in the discussion section. Below are minor points for consideration:

- In the study population section, it is mentioned that a couple of hospitals were not included in the cluster RCT analysis, can you briefly explain why? And were these hospitals included in this analysis?

- Can you please spell out the acronym AMANAT, or add to the acronym list

- Was the first knowledge test undertaken before or after the first simulation training in October 2016? I am assuming before, but think it would be worth being explicit about this.

- You mention that analyses were done on aggregated data, rather than making comparisons in scores on an individual level. Was this not possible at all, even on a smaller sub-set of people who took part, who were still in post at the end? Appreciate for staff who have left positions etc. they would not be included in an individual analysis, but wondered whether you might be able to analyse a sub-set?

- In the “general communication” questions, can you give a brief example of what kind of things are included?

- I think it is worth briefly commenting upon the impact of the Kenyan strike and the potential impact this had on the training itself and subsequent scores. If the training was repeated once the strike was over (due to many staff leaving) in 2017, this could impact interpretation of the results given that, in this setting, additional training was undertaken – if I have understood the manuscript correctly.

Reviewer #2: Thanks for the opportunity to review this article. Please see attached for comments. I believe with revision, this paper could be a valuable addition to the literature and commend the authors for writing it.

6. PLOS authors have the option to publish the peer review history of their article (what does this mean?). If published, this will include your full peer review and any attached files.

**Do you want your identity to be public for this peer review?** For information about this choice, including consent withdrawal, please see our Privacy Policy.

Reviewer #1: No

Reviewer #2: No

---

## [Decision Letter · Decision Letter 1]

23 Mar 2023

PGPH-D-22-01386R1

Simulation and team training to improve preterm birth knowledge, evidence-based practices, and communication skills in midwives in Kenya and Uganda: findings from a pre- and post-intervention analysis

Dear Dr. Miller,

Thank you for submitting your manuscript to PLOS Global Public Health. After careful consideration, we feel that it has merit but does not fully meet PLOS Global Public Health’s publication criteria as it currently stands. Therefore, we invite you to submit a revised version of the manuscript that addresses the points raised during the review process.

Most previous reviewer comments have been adequately addressed. However, the second reviewer has highlighted outstanding concerns and areas where further clarification would aid interpretation of methods and results. In particular, the concerns regarding balance between consistent labeling of a branded intervention model and contextualizing the intervention approach in the broader evidence base are valid and merit further attention.

We look forward to receiving your revised manuscript.

Kind regards,

Hannah Tappis, DrPH, MPH

Academic Editor

Journal Requirements:

2. Please include a complete copy of PLOS’ questionnaire on inclusivity in global research in your revised manuscript. Our policy for research in this area aims to improve transparency in the reporting of research performed outside of researchers’ own country or community. The policy applies to researchers who have travelled to a different country to conduct research, research with Indigenous populations or their lands, and research on cultural artefacts. The questionnaire can also be requested at the journal’s discretion for any other submissions, even if these conditions are not met.  Please find more information on the policy and a link to download a blank copy of the questionnaire here: https://journals.plos.org/globalpublichealth/s/best-practices-in-research-reporting. Please upload a completed version of your questionnaire as Supporting Information when you resubmit your manuscript.

3. Please amend your detailed Financial Disclosure statement. This is published with the article. It must therefore be completed in full sentences and contain the exact wording you wish to be published.

4. We do not publish any copyright or trademark symbols that usually accompany proprietary names, eg  ©, ®, ™  (e.g. next to drug or reagent names). Please remove all instances of trademark/copyright symbols throughout the text, including ® and ™ on pages 4, 5 and 14.

Additional Editor Comments (if provided):

We urge authors to further consider reviewer concerns regarding balance between consistent labeling of a branded intervention model and contextualizing the intervention approach in the broader evidence base. Once the study rationale, including differences between PRONTO’s branded simulation and team training and other capacity building approaches, is well established in the Introduction and intervention components detailed in the Methods, there is no need to continually refer to PRONTO throughout the manuscript. In the Discussion, omission of the word ‘PRONTO’ or reference to ‘this adapted training program’, ‘interactive simulation and team training’, etc. would be sufficient. Emphasis should be on the characteristics of the intervention being evaluated, not on the organization that designed or implemented it.

While there is cursory mention of ‘many programs that emphasize skill building’ and ‘other approaches to improve pre-term knowledge and skills in low-resource healthcare facilities’ with additional references to studies of related interventions, discussion of alternative curricula and approaches remains limited. Further reflection on how this model compares with other promising capacity building approaches in maternal and newborn health (or other clinical service areas) would improve objectivity, and underscore this study’s contribution to knowledge and practice.

Reviewers' comments:

Reviewer's Responses to Questions

**Comments to the Author**

1. If the authors have adequately addressed your comments raised in a previous round of review and you feel that this manuscript is now acceptable for publication, you may indicate that here to bypass the “Comments to the Author” section, enter your conflict of interest statement in the “Confidential to Editor” section, and submit your "Accept" recommendation.

Reviewer #1: All comments have been addressed

Reviewer #2: (No Response)

2. Does this manuscript meet PLOS Global Public Health’s publication criteria? Is the manuscript technically sound, and do the data support the conclusions? The manuscript must describe methodologically and ethically rigorous research with conclusions that are appropriately drawn based on the data presented.

Reviewer #1: Yes

Reviewer #2: Partly

3. Has the statistical analysis been performed appropriately and rigorously?

Reviewer #1: I don't know

Reviewer #2: Yes

4. Have the authors made all data underlying the findings in their manuscript fully available (please refer to the Data Availability Statement at the start of the manuscript PDF file)?

Reviewer #1: No

Reviewer #2: Yes

5. Is the manuscript presented in an intelligible fashion and written in standard English?

Reviewer #1: Yes

Reviewer #2: Yes

6. Review Comments to the Author

Reviewer #1: Thank you for addressing comments - the manuscript is improved.

Reviewer #2: See attached

7. PLOS authors have the option to publish the peer review history of their article (what does this mean?). If published, this will include your full peer review and any attached files.

**Do you want your identity to be public for this peer review?** For information about this choice, including consent withdrawal, please see our Privacy Policy.

Reviewer #1: No

Reviewer #2: No

---

## [Editor Report · Decision Letter 2]

11 May 2023

Simulation and team training to improve preterm birth knowledge, evidence-based practices, and communication skills in midwives in Kenya and Uganda: findings from a pre- and post-intervention analysis

PGPH-D-22-01386R2

Dear Ms. Miller,

We are pleased to inform you that your manuscript 'Simulation and team training to improve preterm birth knowledge, evidence-based practices, and communication skills in midwives in Kenya and Uganda: findings from a pre- and post-intervention analysis' has been provisionally accepted for publication in PLOS Global Public Health.

Best regards,

Hannah Tappis, DrPH, MPH

Academic Editor